# Automatic Text Summarization for Hindi Using Real Coded Genetic Algorithm

**Arti Jain** [1,*], **Anuja Arora** [1] , **Jorge Morato** [2] , **Divakar Yadav** [3] and **Kumar Vimal Kumar** [1]

1    Department of CSE, Jaypee Institute of Information Technology, Noida 201309, India;
     anuja.arora29@gmail.com (A.A.); vimalkumar.k@gmail.com (K.V.K.)
2    Computer Science, Universidad Carlos III de Madrid, 28911 Leganes, Spain; jmorato@inf.uc3m.es
3    Department of CSE, NIT Hamirpur, Hamirpur 177005, India; divakar.yadav0@gmail.com
*    Correspondence: ajain.jiit@gmail.com; Tel.: +91-9313519476

**Featured Application: This paper provides applicability of the Real Coded Genetic Algorithm to the Natural Language Processing Task, i.e., Text Summarization. The purpose of text summarization is to reduce an extensive document into a concise format such that the essence of the content is retained. By doing so, users can utilize the summarized document for vivid applications such as Question Answering, Machine Translation, Fake News Detection, and Named Entity Recognition to name a selected few.**

**Abstract:** In the present scenario, Automatic Text Summarization (ATS) is in great demand to address the ever-growing volume of text data available online to discover relevant information faster. In this research, the ATS methodology is proposed for the Hindi language using Real Coded Genetic Algorithm (RCGA) over the health corpus, available in the Kaggle dataset. The methodology comprises five phases: preprocessing, feature extraction, processing, sentence ranking, and summary generation. Rigorous experimentation on varied feature sets is performed where distinguishing features, namely- sentence similarity and named entity features are combined with others for computing the evaluation metrics. The top 14 feature combinations are evaluated through Recall-Oriented Understudy for Gisting Evaluation (ROUGE) measure. RCGA computes appropriate feature weights through strings of features, chromosomes selection, and reproduction operators: Simulating Binary Crossover and Polynomial Mutation. To extract the highest scored sentences as the corpus summary, different compression rates are tested. In comparison with existing summarization tools, the ATS extractive method gives a summary reduction of 65%.

**Keywords:** automatic text summarization; extractive summary; feature set; Hindi language; Hindi health data; named entity; real coded genetic algorithm; ROUGE metric; summarization tool

## 1. Introduction

Automatic Text Summarization (ATS) [1,2] is a process to generate a summary while preserving the essence, by eliminating irrelevant or redundant content from the text. ATS provides vital information in a much shorter version, usually reduced to less than half of the length of the input text. It remedies the challenge of information overload and helps in information retrieval tasks. ATS provides concise information with reduced redundancy [3] in an effective manner related to news articles [4], emails, official government documents, and many more. In generality, ATS utilizes either an extractive summary [5] or an abstractive summary [6]. An extractive summary is generated while selecting essential sentences from the given textual document. The sentence selection criteria are based on the text's statistical parameters and linguistic features to combine those sentences into the final summary. On the other hand, an abstractive summary is generated by considering into the more profound understanding of semantics for the given textual document. It uses diversified linguistic features to examine and interpret the text and generate new sentences.

In this research, we have worked upon ATS for the Hindi language using an extractive strategy over the Hindi Health Data (HHD) corpus, which is available in the Kaggle datasets (Section 4.1). Hindi is written in the Devanagari script [7] and is an official language of India along with English. Although much work is available on the English text summarization [1,5,6,8,9], comparatively, lesser research is performed in the case of the Hindi language. Hindi serves as a native language for most people living in the north-central states of India- Himachal Pradesh, Uttarakhand, Haryana, Delhi, Uttar Pradesh, Chhattisgarh, Madhya Pradesh, Jharkhand, Bihar, and Rajasthan. It is the third most spoken language globally [10], the mother tongue of 343.0 million people, and the second language for 258.3 million people. Moreover, Hindi is spoken in many countries outside India (Trinidad and Tobago, Guyana, Fiji, Suriname, Mauritius, and Singapore). The extensive Hindi speaking population has motivated the focus of this research on this language.

An extractive summarization is supported by pre-processing, feature extraction, and processing phases. In this study, the pre-processing phase includes sentence segmentation, word tokenization, stemming, Part-of-Speech (POS) tagging, and stop-words removal. The feature extraction phase includes eight features: sentence paragraph position, numerical data, sentence length, keywords within a sentence, sentence similarity, Named Entities (NEs) [11,12], English-Hindi words within a sentence, and Term Frequency (TF)-Inverse Sentence Frequency (ISF). These features influence the importance of sentences using the weighted-learning concept so that final sentence scores are calculated using feature weights.

The size and complexity of generated features is a problem for ATS processing of the Hindi language. To reduce the dimensionality problem of generated feature data, an optimization algorithm is requisite which can explore the features search space to exploit the best feature set for text summarization. Although automatic text summarization for the English language has worked on numerous optimization algorithms, Hindi Language automatic text summarization needs systematic consideration. The generated features using the mentioned steps above contain real values. To deal with real values of generated features, Sarkar et al. [13] suggested and validated a real-valued Genetic Algorithm (GA) for Natural Language Processing (NLP) [14] research problems. The idea to apply a real coded genetic algorithm is taken from Goldberg et al. [15]. The processing phase considers the Real Coded Genetic Algorithm (RCGA), which search capability explores the best feature weights and differentiates significant features. The feature combinations have been experimented with over the test dataset for different compression rates. Then, the best-scored sentences are picked up and added to the final summary, and the generated summary is compared with other existing tools. The research contributions of the proposed approach are as follows:

- To introduce a Real Coded Genetic Algorithm aimed at automatic text summarization for the Hindi language.
- To work with extensive and novel Hindi language features that generate more accurate text summarization than generic features.
- To compute better and faster convergence for text summarization using Simulating Binary Crossover and Polynomial Mutation in RCGA.
- To evaluate summarization results using Recall-Oriented Understudy for Gisting Evaluation (ROUGE) metrics.

The rest of the paper is organized as follows. Section 2 discusses the related work. Literature is studied in different perspectives to understand the research gaps and build a successful automatic text summarization system. Section 3 provides the detailed methodology which covers text pre-processing, feature extraction, automatic text generation processing, sentence ranking, and summary generation phase. Section 4 illustrates an experimental setup which includes dataset, evaluation metrics and results. Section 5 discusses the findings of this study. Section 6 provides research directions for the future.

## 2. Related Work

This section is subdivided into four subsections: types of an extractive method, genetic algorithm for summarization, Automatic Text Summarization (ATS) for the Hindi language, and existing tools for text summarization, respectively.

### 2.1. Types of Extractive Methods

ATS-based extractive methods [16,17] are classified as statistical, linguistic, and hybrid approaches.

- Statistical methods: Statistical-based text summarization [18–20] relies on the statistical distribution of specified features without understanding of the whole document. For the selection of a sentence in a document summary, weight is assigned to the sentence as per its level of importance. Examples of statistical features are title words, sentence length, thematic topic, etc.
- Linguistic methods: Linguistic-based text summarization [19,21] relies on deep linguistic knowledge to analyze sentences and then decide which sentences to select. Sentences that contain proper nouns and pronouns have a greater chance to be included in the document summary. Examples of linguistic features are named entity features, sentence-to-sentence similarity features, etc.
- Hybrid methods: Hybrid-based text summarization [22] relies on optimizing the best of both the previous methods. It incorporates the combination of statistical and linguistic features for a meaningful and short summary.

### 2.2. Genetic Algorithm for Summarization

The Genetic Algorithm [23,24] is observed as an optimization function with a wide range of application domains. GA is applied to text clustering [25], query path optimization [26], pattern recognition [27], intrusion detection [28], and so on. Another wide usage of genetic algorithms is for the extractive text-based summarization task over available datasets for different languages while using statistical and linguistic features and to evaluate summarization measures (Table 1).

**Table 1.** Genetic algorithm based extractive text summarization task.

| Source | Dataset(s) | Language(s) | Features Extraction | Evaluation Measures |
|---|---|---|---|---|
| Litvak et al., 2010 [29] | DUC 2002—533 news articles; Haaretz newspaper—50 news articles | English Hebrew | 31 sentence scoring metrics | ROUGE-1 English: 44.61% ROUGE-1: 59.21% |
| Suanmali et al., 2011 [30] | DUC 2002—100 documents | English | title feature, sentence length, term weight, sentence position, sentence-to-sentence similarity, proper noun, numerical data, thematic word | Precision: 49.80% Recall: 44.64% F-score: 46.62% |
| Abuobieda et al., 2012 [31] | DUC 2002—100 documents | English | title feature, sentence length, sentence position, numerical data, thematic words | ROUGE-1 |
| García-Hernández and Ledeneva, 2013 [32] | DUC 2002—567 news articles | English | n-gram, frequency of words, sentence position | F-score (48.27%) |
| Thaokar and Malik, 2013 [33] | - | Hindi | statistical features: TF-ISF, sentence length, sentence position, numerical data, sentence-to-sentence similarity, title feature; linguistic features: SOV qualification, subject similarity | Precision Recall |
| Kadam et al., 2015 [34] | - | Hindi | statistical features: TF-ISF, sentence length, sentence position, numerical data, sentence-to-sentence similarity, title word; linguistic features: proper noun, thematic words | Suggest ROUGE (unspecified) |

**Table 1.** *Cont.*

| Source | Dataset(s) | Language(s) | Features Extraction | Evaluation Measures | | | |
|---|---|---|---|---|---|---|---|
| Pareek et al., 2017 [35] | Dainik Bhaskar, Patrika, Zee news- 15 text documents with categories: sports, Bollywood, politics, science, history | Hindi | statistical features: TF-ISF, sentence length, sentence position, numerical data, sentence to sentence similarity, title word, adjective feature; linguistic features: SOV qualification, subject similarity | Com. Rate (%)<br>25<br>50<br>60 | Precision (%)<br>69<br>73<br>78 | Recall (%)<br>62<br>67<br>79 | Accuracy (%)<br>67<br>71<br>80 |
| Vázquez et al., 2018 [36] | DUC01—309 news articles; DUC02—567 news articles | English | similarity to title, sentence position, sentence length, term length, coverage | DUC<br>2001<br>2002 | | ROUGE-1 (%)<br>45.058<br>48.423 | ROUGE-2 (%)<br>19.619<br>22.471 |
| Simón et al., 2018 [37] | DUC01—309 news articles; DUC02—567 news articles | English | term frequency (document and summary), sentence position | DUC<br>2001<br>2002 | | ROUGE-1 (%)<br>37.39<br>41.06 | ROUGE-2 (%)<br>38.39762<br>40.6138 |
| Anh et al., 2019 [38] | CNN/DailyMail | English | TF-ISF, similarity to topic, sentence length, proper noun, sentence position | Precision (%)<br>84.3 | | Recall (%)<br>48.3 | F-measure (%)<br>58.0 |
| Hernández-Castañeda et al., 2020 [39] | DUC02 and TAC11 | English Arabic, Czech, French, Greek, Hebrew, and Hindi | TF-IDF, one-hot encoding, latent dirichlet allocation, Doc2Vec | ROUGE-1 | | English<br>Arabic<br>Czech<br>French<br>Greek<br>Hebrew<br>Hindi | 0.48681<br>0.33913<br>0.43643<br>0.49841<br>0.32770<br>0.30576<br>0.11351 |
| Chen et al., 2021 [40] | CNN/DailyMail | English | vocabulary set | Training Size<br>100<br>50 | | Vocab Size<br>90,000<br>90,000 | ROUGE-1 Score<br>23.59<br>22.6 |
| Tanfouri et al., 2021 [41] | EASC—153 articles and Multilingual Pilot Corpus 2013 | Arabic | - | EASC<br>Multilingual | | ROUGE-1<br>0.41<br>0.16 | ROUGE-2<br>0.30<br>0.029 |
| Khotimah and Girsang, 2022 [42] | IndoSum—60 documents with 6 different topics | Indonesian | | Com. Rate (%)<br>10<br>20<br>30 | Precision (%)<br>48.9<br>38.7<br>33.0 | Recall (%)<br>40.6<br>53.9<br>64.0 | F-score (%)<br>42.6<br>43.4<br>42.1 |

It has been observed in the literature study phase that text summarization research for Hindi or other languages generally uses a genetic algorithm [43]. Feature generation discretizes the real-valued features and this could be the reason for achieving poor performance in the existing works, whereas actual quantitative feature values can help in improving performance [44]. Henceforth, a real-valued/coded genetic algorithm is explored and applied for text summarization of the Hindi Language by making use of generated features.

### 2.3. ATS for the Hindi Language

Few studies have been carried out for automatic text summarization for the Hindi language. Thaokar and Malik [33] summarize the Hindi textual documents using GA based extraction method. They analyze statistical and linguistic features to extract sentences and use Hindi Wordnet [45]—a Hindi lexical resource for checking the word order of the Hindi sentences. Their method maximizes the theme coverage while minimizing content redundancy. Anitha et al. [22] propose summarization for the Hindi text while combining fuzzy and neural networks to generate sentence scores. In this study global search optimization is combined with neural networks to optimize the weight criteria and use the hybrid score for the generated summary. Their approach computes precision (90%) and recall (88%) at a 20% compression rate. Kadam et al. [34] compare the sentence extraction-based ATS approach for Hindi using neural networks and genetic algorithms. Kumar et al. [46] propose a graph-based technique for the Hindi text summarization. Their system achieves an average precision (79%), recall (69%), and F-measure (70%), respectively. Desai and Shah [18] discuss ATS for Hindi using a supervised learning technique while considering

features, e.g., overall sentence position, length of sentence, and occurrence of inverted commas for the generation of the textual summary. Gupta and Garg [21] mention a rule-based approach for the Hindi text summarization. They use deadwood removal to generate the summary, where deadwood refers to a word or phrase which carries no meaning in the summarization task. Their system gives 96% accurate results and a 60–70% compression rate when tested on 30 different documents. However, their system does not consider the semantics of the text. Sargule and Kagalkar [47] propose the Bernoulli model of randomness for the Hindi text summarization. They develop a graph-based ranking algorithm that computes an informative measure for each document term and generates indexing values for each word within the document. These indexes are used to select words for the summary. Desai and Shah [18] discuss Support Vector Machine (SVM) based single document summarization for the Hindi language. They divide the sentences into 4 categories—most influential, important, less important, and insignificant. The experiments are carried out on news stories from categories such as Bollywood, politics, and sports. Their results show 72% accuracy at a compression ratio of 50%, and 60% accuracy at a compression ratio of 25%. Giradkar et al. [48] propose a back-propagation based on multiple Hindi textual document summarization systems. They use features such as sentence length, the position of the sentence, the similarity of the sentence, and others. Among the 70 chosen documents, they use 50 documents to train the network, while the remaining 20 documents are for testing purposes. Dalal and Malik [49] propose bio-inspired computing for the Hindi text summarization. They use semantic graphs and particle swarm optimization algorithms. The semantic graph captures the semantic structure of the textual document. The particle swarm optimization searches for the optimal solution irrespective of the large dimensional space. They achieve precision (42.86%), recall (60%), and F-measure (50.01%), respectively. Pareek et al. [35] work with a genetic algorithm for the Hindi ATS system while collecting online news articles from Dainik Bhaskar, Patrika, and Zee news where 15 text documents are of categories like Bollywood, politics, sports, science, and history. They used nine features; among them, 6 are statistical features (TF-ISF, sentence length, sentence position, numerical data, sentence to sentence similarity, and title word), and 3 are linguistic features (subject-object-verb qualification, subject similarity, and adjective feature). In this work evaluation measures are computed at different compression rates of 25%, 50%, and 60%, respectively. Rani and Lobiyal [50] develop an extractive lexical knowledge-rich topic modeling text summarizing approach. The tagged Latent Dirichlet Allocation (LDA) modeling is applied over 114 Hindi novels including short stories from Munshi Premchand's stories. The system performs preferably to the baseline algorithms for 10–30% compression ratios and given evaluation metrics. However, semantic features are not taken into consideration and hence, results are somewhat declined when compared with baseline systems for different compression rates. Yadav et al. [1] provide a comprehensive review of automatic text summarization methods. They discuss that automatic text summarization is difficult and demanding for the Hindi language and is still an unsolved topic due to lack of corpus and insufficient processing tools.

### 2.4. Tools for Text Summarization

Certain tools are widely being used for automatic text summarization. Some of them are discussed here:

- Newsblaster: Newsblaster [51] is a summarizing system which is developed at Columbia University. It generates news updates on a daily basis by scraping data from different news websites, filtering out news from non-news aspects such as advertisements, and combining them to generate a summary for each event. The evaluation measures for the Newsblaster multilingual summarization system [52] are computed for the English, Russian and Japanese languages (Table 2).

**Table 2.** Newsblaster performance for different languages.

| Language | No. of Articles | Precision | Recall |
|---|---|---|---|
| English | 353 | 89.10% | 90.70% |
| Russian | 112 | 90.59% | 95.06% |
| Japanese | 67 | 89.66% | 100% |

- Tool4Noobs: Tool4Noobs [53] is another tool to summarize text documents while setting different parameters (Figure 1). These parameters are threshold value or number of lines, minimum sentence length, minimum word length, number of keywords, etc. The evaluation measures for Tool4Noobs are computed as precision (61.4%), recall (63.2%), and F-Measure (62.2%), respectively.

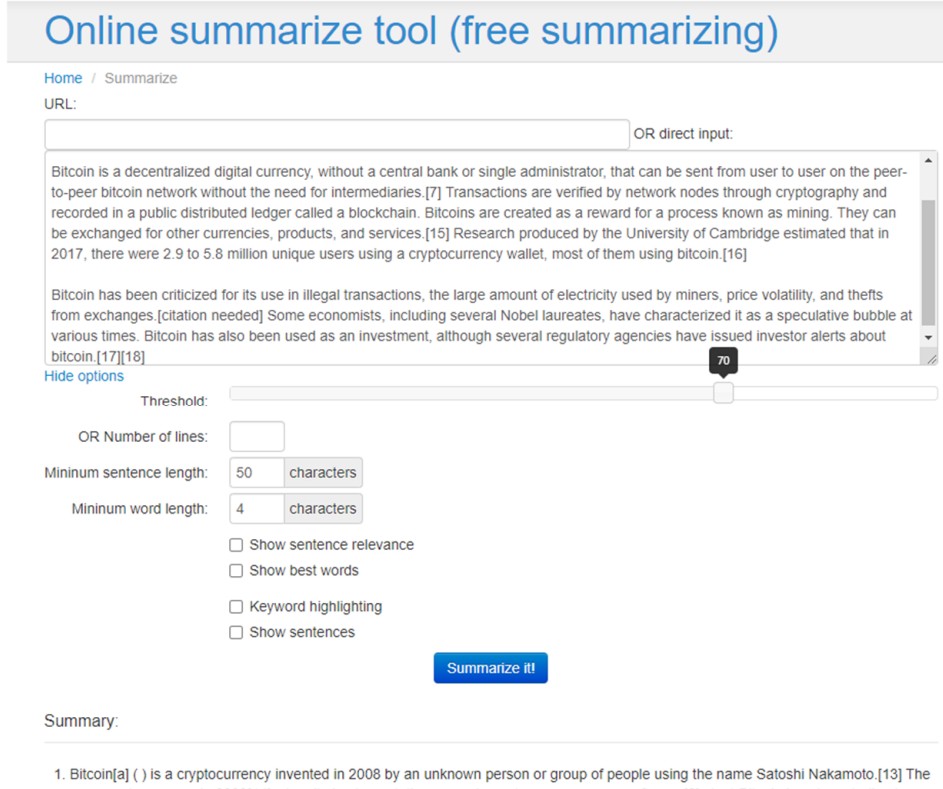

**Figure 1.** Tool4Noobs summary tool.

- Free Summarizer: Free Summarizer [54] is a free service, an online tool developed to summarize mono-lingual texts quickly. This tool generates an extractive summary of generic types from a single text document and is widely used by various users worldwide.
- SMMARY: SMMRY [55] is another online tool that accomplishes text summarization while ranking sentences based on relevance, selecting keywords to focus on a particular topic, and removing unnecessary clauses (Figure 2).
- M-HITS: M-HITS [56] is a Hindi text summarizer that incorporates supervised learning algorithms such as Support Vector Machine (SVM), Random Forest (RF), AdaBoost, Gradient Boost, K-Nearest Neighbor (KNN), Logistic Regression (LR), Extremely Randomized Trees, along with graph-based similarity to extract main text chunks using various statistical features. These features are- cue words, bigrams, topic features, sentence position, proper nouns, unknown words, and TF-IDF. This tool enables ATS even without requiring a deep understanding of the text.

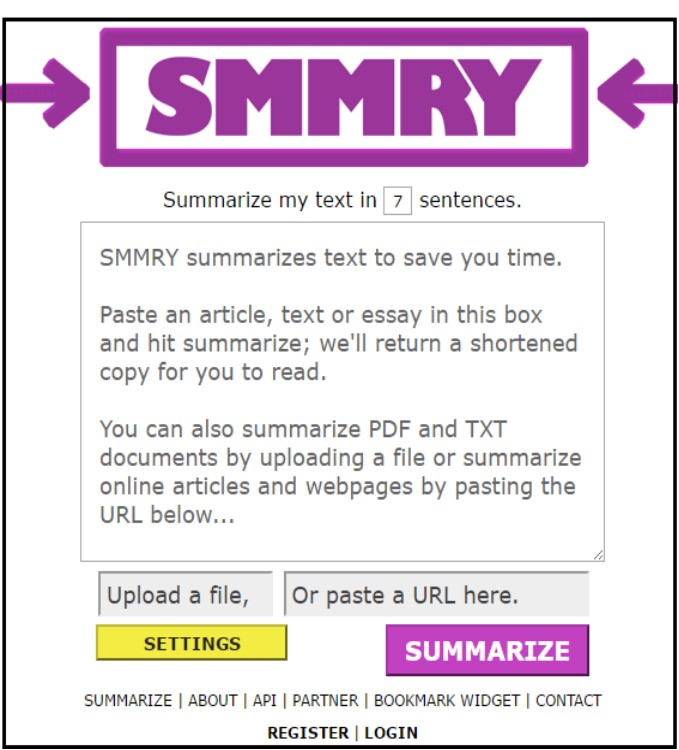

**Figure 2.** SMMRY tool.

### 3. Proposed Methodology

The proposed Automatic Text Summarization (ATS) methodology for the Hindi language is based upon an extractive method that includes five prime phases- preprocessing phase, feature extraction phase, processing phase, sentence ranking phase, and summary generation phase. Each of these phases is detailed here.

Hindi Health Data (HHD) corpus from the Kaggle dataset (Section 4.1) is taken as input and undergoes the pre-processing phase. The pre-processing phase comprises sentence segmentation, word tokenization, stemming, POS tagging, and stop-word removal. The cleaned HHD corpus is then passed through the feature extraction phase. The feature extraction phase comprises extracting features such as sentence paragraph position, numerical data, length of sentence, keywords, sentence similarity, named entities, English-Hindi words, and TF-ISF. The HHD corpus then undergoes the processing phase where usage of real coded genetic algorithms is performed. RCGA comprises an initial population of strings/chromosomes from the HHD corpus, followed by the desired subset of chromosomes selection to proceed for the next generation while choosing an appropriate fitness function. The selection of the best chromosomes is performed via crossover and mutation operators, which are applied to suitable extracted features. The entire RCGA process is repeated until a stopping criterion is reached. After the RCGA process is over, sentence ranking is performed over the extracted HHD features with their appropriate weights, and hence the score of the sentences is considered, which generates a summary of the HHD corpus.

#### 3.1. Pre-Processing Phase

The preprocessing phase is performed to clean the HHD corpus, including sentence segmentation, word tokenization, stemming, POS tagging, and stop-word removal. Each of these preprocessing modules is discussed here in detail.

### 3.1.1. Sentence Segmentation

In sentence segmentation, the HHD corpus is split into its constituent sentences while taking into account the exemplified boundary constraints such as "पूर्ण-विराम " (poorn-viram/full-stop) or "प्रश्न-चिन्ह " (prashn-chinh/question-mark).

### 3.1.2. Word Tokenization

In word tokenization, the HHD corpus is divided into several tokens while identifying special symbols, such as space, comma, colon, semi-colon, dash, hyphen, closing bracket, quotes, and exclamation mark. For example, consider the phrase "गुर्दे की पथरी के कारण" (gurde kee patharee ke kaaran/causes of kidney stones) which comprise of five tokens.

### 3.1.3. Stemming

In stemming, the HHD corpus contains root-words which are identified while removing more than 50 suffixes [57] to obtain words with common origin and to improve effectiveness. For example, root word of "लाभदायक" (laabhadaayak/profitable), and "लाभकारी" (laabhakaaree/beneficial) is "लाभ" (laabh/benefit).

### 3.1.4. POS Tagging

In POS tagging, the HHD corpus is assigned the POS tags which are based on the Hindi grammar [7], such as "संज्ञा" (sangya/noun), "सर्वनाम" (sarvanaam/pronoun), "क्रिया" (kriya/verb), "विशेषण" (visheshan/adjective), "क्रियाविशेषण" (kriya visheshan/adverb), etc. Further, the POS tagged corpus eases the extraction of TF-ISF and sentence similarity features. TF-ISF feature extracted from an untagged corpus does not carry relevant information required for the summarization process. Moreover, the sentence similarity contains syntactic similarity of the corpus words, calculated using the tagged corpus.

### 3.1.5. Stop-Words Removal

In stop-word removal, the HHD corpus does not further process the words with little lexical meaning, such as determiners, conjunctions, articles, prepositions, and pronouns. A list of 225 Hindi stop-words is considered from GitHub [58]. For example: "अपना", "इन", "जैसे", "तो", "था".

At the end of the preprocessing phase, the cleaned HHD corpus is prepared for further analysis.

### 3.2. Feature Extraction Phase

In the feature extraction phase [59], the cleaned HHD corpus analysis is initiated for the ATS purpose. In this phase, each Hindi sentence is represented as a weighted feature vector within a range of 0 to 1 and is used to rank the sentences. At present, eight features are used, each of them is detailed as follows.

### 3.2.1. Sentence Paragraph Position ($F_{sp}$)

The position of a sentence within HHD paragraphs has its significance. For example, the start of the paragraph may comprise a sentence with a higher probability of occurring within the summary as it conveys the document theme. Mathematically, sentence paragraph position value is calculated using Equation (1).

$$s_i = \frac{n - i}{n} \tag{1}$$

where

$n$: no. of sentences in paragraph;
$i$: range from 0 to $n$;
$s_i$: $i$th sentence within a paragraph.

### 3.2.2. Numerical Data within Sentence ($F_{nd}$)

The numerical data with HHD sentences represent informative information regarding time: "2बजे" (2 baje/2 o'clock); quantity: "3–4 बार" (3–4 baar/3–4 times); percentage: "12% रोगी" (12% rogee/12% patient); home remedies: 1. "लौंग" (laung/clove), 2. "अखरोट का तेल" (akharot ka tel/walnut oil), etc. Mathematically, numerical data within sentence $s_i$ are calculated using Equation (2).

$$n_d\_s_i = \frac{n\_s_i}{w\_s_i} \tag{2}$$

where

$n\_s_i$: total no. of numeric values in $s_i$;
$w\_s_i$: total no. of words in $s_i$;
$n_d\_s_i$: numeric data within the sentence.

### 3.2.3. Length of Sentence ($F_{ls}$)

The HHD sentences which are either too short or too long in length are not a good choice for the summary. Mathematically, the length of sentence value is calculated using Equation (3).

$$l_n\_s_i = \frac{w\_s_i}{l_g\_s_i} \tag{3}$$

where

$w\_s_i$: total no. of words in $s_i$;
$l_g\_s_i$: total no. of words in the longest sentence;
$l_n\_s_i$: length of $s_i$.

### 3.2.4. Keywords within Sentence ($F_{kw}$)

The identification of HHD keywords serves as an important summarization feature. The HHD keywords are those words/phrases that reflect the corpus, act as indices, and project the core sentiment of the corpus. Their target is to speed up computation abilities and information organization of the system without any human intervention. For example: "रोग" (rog/disease), "लक्षण" (lakshan/symptom), "इलाज" (ilaaj/treatment), "संक्रमण" (sankraman/infection), and "बचाव" (bachaav/rescue). Mathematically, keywords within Hindi sentences are calculated using Equation (4).

$$k_w\_s_i = \frac{k\_s_i}{w\_s_i} \tag{4}$$

where

$k\_s_i$: total no. of keywords in $s_i$;
$w\_s_i$: total no. of words in $s_i$;
$k_w\_s_i$: keywords data within a sentence.

### 3.2.5. Sentence Similarity ($F_{ss}$)

The sentence similarity feature measures the amount of syntactic and semantic similarity between the HHD sentences. For instance, it computes the similarity between $s_i$ and every HHD sentence. To perform it, the sentence similarity is computed as a combination of maximum semantic similarity and word order similarity (or syntactic similarity), as is detailed in Li et al. [60]. The semantic similarity represents the lexical similarity using Latent Semantic Analysis (LSA) [61]. The syntactic similarity [62] represents a relationship between the words by applying the cosine function over the vectors, which are generated using word positions in the different sentences. Both semantic and syntactic information

plays a pivotal role in comprehending the meaning of the HHD sentences. Mathematically, sentence similarity is calculated using Equation (5).

$$sen\_sim(s_a, s_b) = \delta sem\_sim + (1 - \delta)syn\_sim \tag{5}$$

where

$s_a$, $s_b$: $a$th, $b$th HHD sentences;
*sem_sim*: semantic similarity using LSA;
*syn_sim*: syntactic similarity using cosine function;
$sen\_sim(s_a, s_b)$: similarity between the sentences-$s_a$ and $s_b$;
$\delta$: parameter measuring contribution of semantic and syntactic information to overall sentence similarity, $\delta \in [0.5, 1]$.

### 3.2.6. Named Entities within Sentence ($F_{ne}$)

Named Entity (NE) [63,64] refers to any real-world object that is named, viewed as an entity instance, and has physical or abstract existence. In the HHD corpus, NEs comprise the following types: disease, symptom, consumable, and person, detailed in Jain and Arora [65] (Table 3).

**Table 3.** Sample NEs within HHD corpus.

| NE Type | HHD Example(s) | |
|---|---|---|
| Disease | "दमा" | (dama/asthma) |
| | "मधुमेह" | (madhumeh/diabetes) |
| Symptom | "कमजोरी" | (kamajoree/weakness) |
| | "थकान" | (thakaan/fatigue) |
| Consumable | "कालीमिर्च" | (kaaleemirch/black pepper) |
| | "गाजर" | (gaajar/carrot) |
| Person | "चिकित्सक" | (chikitsak/doctor) |
| | "मरीज" | (mareej/patient) |

Mathematically, named entities within Hindi sentences are calculated using Equation (6).

$$n_w\_s_i = \frac{n_e\_s_i}{w\_s_i} \tag{6}$$

where

$n_e\_s_i$: total no. of NEs in $s_i$;
$w\_s_i$: total no. of words in $s_i$;
$n_w\_s_i$: NEs data within the sentence.

### 3.2.7. English Hindi Words within Sentence ($F_{eh}$)

The English words are commonly being used in Hindi sentences. For example, consider the HHD sentence as "प्रोस्टेट कैंसर (Prostate cancer) एक खतरनाक बीमारी है।" (prostate cancer:ek khataranaak beemaaree hai/prostate cancer is a dangerous disease). This sentence contains प्रोस्टेट कैंसर (Prostate cancer) as English–Hindi nouns which are obviously not available in the Hindi WordNet, however, are quite helpful in determining the importance of the sentence. Mathematically, the common English–Hindi word score within the Hindi sentence is calculated using Equation (7).

$$d_{eh}\_s_i = \frac{t_{eh}\_s_i}{l_n\_s_i} \tag{7}$$

where

$t_{eh}\_s_i$: total English-Hindi words in a sentence;

$l_n\_s_i$: length of $s_i$;
$d_{eh}\_s_i$: English-Hindi data within a sentence.

### 3.2.8. TF-ISF ($F_{ti}$)

TF-ISF refers to the term frequency and inverse sentence frequency, respectively. The term frequency is a measure of the occurrence of a term/word in the HHD sentence. It considers every word with equal importance, which is practically improbable. In such a case, the inverse sentence frequency measure is quite useful, predicting the importance of words based on their usage in the HHD sentences. Mathematically, TF-ISF is calculated using Equations (8) and (9), respectively.

$$Tf_i = \frac{fq_i^j}{w\_s_j} \tag{8}$$

$$Isf_i = \log\left(\frac{t_n s}{n_s\_t_i}\right) \tag{9}$$

where

$fq_i^j$: frequency of $i$th word in $j$th sentence;
$w\_s_j$: number of words in sentence $s_j$;
$t_n s$: total number of sentences;
$n_s\_t_i$: number of sentences with $i$th word;
$Tf_i$: term frequency of $i$th word;
$Isf_i$: inverse sentence frequency of $i$th word.

### 3.3. Processing Phase

The features extracted in the feature extraction phase serve as basic elements for the ATS process. The summary quality is sensitive to these features as they impact how the corpus sentences are scored. It thus becomes crucial to realize the importance of each feature based on their weighted scores. To perform it, the genetic algorithm approach is chosen, which optimizes the feature weights using techniques, such as selection, crossover, and mutation.

#### Genetic Algorithm

A Genetic Algorithm (GA) [40–42] is a randomized search technique that runs a function optimizer to explore and present a globally optimal solution for an optimization problem. GA search space parameters can be encoded as strings of binary digits, 0s and 1s, also called chromosomes. GA transforms the initial population of individual chromosomes into a new population based upon the chromosome fitness value and reproduction operators—crossover and mutation. In this paper, ATS features are encoded into chromosomes, and then GA searches for the selection of appropriate features. The crossover and mutation operators accelerate the convergence of GA. The GA process involving Chromosome Selection, Simulating Binary Crossover, and Polynomial Mutation is continued, either for a specified number of generations or until the termination condition is achieved.

In total, 8 features are generated in the former feature extraction phase for automatic text summarization for the Hindi language. The generated features are real values that lie in the range of 0 to 1 and decision variable/generated features need to be used directly. Hence, a real-coded genetic algorithm is requisite instead of a binary-coded genetic algorithm. In the case of RCGA, naive crossover and single crossover might fail. Additionally, the mutation strategy needs modification.

- String Representation: In this research, Chromosome size $C$ (=8) represents the total number of features or length of a chromosome. Each chromosome is a combination of 8 computed feature values. For a specific sentence, each value is in the range of 0 to 1. Based on this fact, consider the chromosome value at $i = 3$ position is 0.37 is the length

of sentence ($F_{ls}$) feature value. So, each feature value for a specific sentence participates in the text summarization process. In a given population, all its chromosomes are initialized in the same manner.

Table 4 exemplifies a chromosome with real coded values of 8 features—Sentence Paragraph Position ($F_{sp}$: 0.40), Numerical Data ($F_{nd}$: 0.39), Length of Sentence ($F_{ls}$: 0.43), Keyword ($F_{kw}$: 0.76), Sentence Similarity ($F_{ss}$: 0.21), Named Entity ($F_{ne}$: 0.34), English–Hindi words ($F_{eh}$: 0.73), and Term Frequency-Inverse Document Frequency ($F_{ti}$: 0.81). It is used for classifier construction.

**Table 4.** Example of chromosome from the initial population.

| $F_{sp}$ | $F_{nd}$ | $F_{ls}$ | $F_{kw}$ | $F_{ss}$ | $F_{ne}$ | $F_{eh}$ | $F_{ti}$ |
|------|------|------|------|------|------|------|------|
| 0.40 | 0.39 | 0.43 | 0.76 | 0.21 | 0.34 | 0.73 | 0.81 |

- Fitness Function: In the genetic algorithm, the fitness function is a unit measure that determines a chromosome that leads to the best solution among a pool of chromosomes and, hence, has its chance to be chosen as the next generation chromosome. In this research, for each input, maximum value of each $C$ is 1 and minimum is 0 where $C$ represents a number of features, and each $C$ is a real value. The topmost chromosome having the highest recall is selected, and the fitness function is defined using Equation (10).

$$F(s) = \sum_{j=1}^{C} f_j(s) \tag{10}$$

where

$f_j(s)$: $j$th feature of the sentence;
$C$: total number of features;
$F$: fitness function.

- Selection of Best Chromosome: The selection operator determines which individual chromosomes can survive and continue to the next generation. In this research, the top two chromosomes are chosen as parents for the new generation since they give the highest recall measure through the fitness function. For the selection of parents, the most frequently used selection method—the roulette wheel method [66] is chosen, which gives a chance to all the chromosomes without rejecting any of them. In this selection strategy, the whole population is partitioned through several individuals, where each sector of the roulette wheel represents an individual. The proportion of individual fitness to total fitness of the entire population decides—the area of the sector for the individual and the probability of the individual being selected for the next generation. So, at first, it calculates the sum of fitness values of all the chromosomes, i.e., cumulative fitness of the entire population, and then calculates the probability of each chromosome using Equation (11).

$$P(s) = \frac{f_j(s)}{\sum_{j=1}^{C} f_j(s)} \tag{11}$$

where

$f_j(s)$: $j$th feature of the sentence;
$C$: total number of features;
$P$: the probability of a chromosome.

- Crossover: The binary-coded genetic crossover operations cannot be used in real coded GA. Deb and Agrawal [67] have developed a real coded crossover technique, i.e., Sim-

ulating Binary crossover (SBX). The considered crossover probability $P_c = 0.8$ and distribution index of crossover $\eta_c = 20$. The procedure of SBX crossover is as follows:

- Randomly select a pair of parents $P_a$ and $P_b$ from the mating pool;
- Generate a random number $r$ between 0 and 1;
- If $r \leq P_c$, copy the parent as offspring;
- If $r \geq P_c$, generate a new random number $u$, $u \in [0, 1]$ for each feature;
- Determine spread factor $\beta$ of each variable using Equation (12):

$$\beta = \begin{cases} (2u)^{\frac{1}{\eta_c+1}}, & \text{if } u \leq 0.5 \\ \left(\frac{1}{2(1-u)}\right)^{\frac{1}{\eta_c+1}}, & \text{otherwise} \end{cases} \tag{12}$$

- Generate two offspring $O_a$ and $O_b$ using Equations (13) and (14):

$$O_a = 0.5[(1+\beta)P_a + (1-\beta)P_b] \tag{13}$$

$$O_b = 0.5[(1-\beta)P_a + (1+\beta)P_b] \tag{14}$$

Although generating offspring, if the features of the sentences taken (parents) are distant then the offspring generated will also be widely spread, and for near features parents the offspring will be closer. Impact on generated offspring based on the value of $\beta$ will be as follows.

- Contracting crossover, i.e., $\beta < 1$, offspring are closer;
- $\beta = 1$, offspring will be original parents;
- $B > 1$, offspring are far.

For the chromosome presented in Table 4, offspring generation using SBX crossover operation are presented in Table 5 is as follows—Values of $F_{sp}$ (Sentence Paragraph Position), $F_{nd}$ (Numerical Data within Sentence), $F_{ls}$ (Length of Sentence), $F_{ss}$ (Sentence Similarity) and $F_{ne}$ (Named Entities within Sentence) features is $\leq 0.5$, so spread factor $\beta = (2u)^{\frac{1}{\eta_c+1}}$. Additionally, values of $F_{kw}$ (Keywords within Sentence), $F_{eh}$ (English–Hindi Words within Sentence), and $F_{ti}$ (TF-ISF) is >0.5, so spread factor $\beta = \left(\frac{1}{2(1-u)}\right)^{\frac{1}{\eta_c+1}}$ as mentioned in Equation (12).

**Table 5.** Generated offspring after crossover operation with feature values.

| Feature | $F_{sp}$ | $F_{nd}$ | $F_{ls}$ | $F_{kw}$ | $F_{ss}$ | $F_{ne}$ | $F_{eh}$ | $F_{ti}$ |
|---|---|---|---|---|---|---|---|---|
| Chromosome | 0.40 | 0.39 | 0.43 | 0.76 | 0.21 | 0.34 | 0.73 | 0.81 |
| Offspring (population size = 4) | 0.956 | 0.951 | 0.97 | 1.16 | 0.84 | 0.925 | 1.13 | 1.21 |

- Mutation: Deb and Agarwal [68] advised polynomial mutation which has been used for variation in population for the research work. The considered mutation probability $P_m = 0.2$ and distribution index of mutation $\eta_m = 20$. High crossover probability and low mutation probability is taken for better outcome. The step of polynomial mutation are as follows:

- Generate a random number $u$ between 0 and 1;
- If $u \geq P_m$, then no change in population;
- If $u < P_m$, new random number $r \in [0, 1]$ corresponding to each feature;
- Determine $\delta$ of each variable using Equation (15):

$$\delta = \begin{cases} (2r)^{\frac{1}{\eta_m+1}} - 1, & \text{if } r < 0.5 \\ 1 - 2(1-r)^{\frac{1}{\eta_m+1}}, & \text{if } r \geq 0.5 \end{cases} \tag{15}$$

- Modify offspring using Equation (16):

$$Offspring_{new} = P_a + (ub - lb) * \delta \tag{16}$$

$P_a$ is the parent, $ub$ and $lb$ are the upper and lower bound values of each feature in the chromosome. In our case, lower bound $lb$ is 0 and upper bound $ub$ is 1, as all feature values lie in the range of 0 and 1.

For the offspring presented in Table 5, mutated offspring generation using Polynomial Mutation operation are presented in Table 6 is as follows—Generate random number $r$ and consider random value $r$ of $F_{kw}$, $F_{eh}$, and $F_{ti}$ are < 0.5, i.e., $\delta = (2r)^{\frac{1}{\eta_m+1}} - 1$, $if\ r < 0.5$ and for all other offspring's $r$ value is > 0.5, so, $\delta = 1 - (2(1-r))^{\frac{1}{\eta_m+1}}$, $if\ r \geq 0.5$, as mentioned in Equation (15).

**Table 6.** Generated mutated offspring after polynomial mutation operation with feature values.

| Feature | $F_{sp}$ | $F_{nd}$ | $F_{ls}$ | $F_{kw}$ | $F_{ss}$ | $F_{ne}$ | $F_{eh}$ | $F_{ti}$ |
|---|---|---|---|---|---|---|---|---|
| Chromosome | 0.40 | 0.39 | 0.43 | 0.76 | 0.21 | 0.34 | 0.73 | 0.81 |
| Offspring SBX (population size = 4) | 0.956 | 0.951 | 0.97 | 1.16 | 0.84 | 0.925 | 1.13 | 1.21 |
| Offspring (mutation) | 0.385 | 0.371 | 0.43 | 0.183 | 0.21 | 0.315 | 0.177 | 0.19 |

Now fitness function value will be calculated for this mutated offspring and the entire RCGA process is continued until the fitness value of the chromosomes within the population either converges, or a fixed number of generations is reached, i.e., until the GA process ceases to improve.

### 3.4. Sentence Ranking Phase

In order to rank the corpus sentences, the RCGA process selects the best chromosome after a specific number of generations. The Euclidean distance measure is applied to evaluate the distance between the sentence score and the fittest chromosome. Based on ranking, sentences are sorted in ascending order of their distance value.

### 3.5. Summary Generation Phase

At the end of the ATS process, depending upon the compression rate—a set of the highest scored sentences from the sentence ranking phase are extracted as a corpus summary and in the original order as in the HHD corpus.

### 4. Experimental Setup

This section describes the chosen health dataset, evaluation metrics, and results obtained during the experimentation.

### 4.1. Dataset

The Hindi Health Data (HHD) corpus is publicly available at the Kaggle dataset [69], which was collated between 2016 and 2018 from Indian websites, namely—Traditional Knowledge Digital Library (TKDL), Ministry of Ayush, University of Patanjali, and Linguistic Data Consortium for Indian Languages (LDC-IL). The corpus serves as a repository in the Hindi language for the health and allied domains towards research activities. The HHD has Unicode Transformation Format (UTF) encoding, mainly in Hindi and a few English phrases. The corpus encloses varied diseases with the following details: disease name, disease description, symptoms and reasons of disease, treatments, and home remedies to cure disease. The corpus comprises approximately 234 pages of MSWord document; 5236 paragraphs; 9496 lines; 105,050 words; 411,462 characters (no spaces); and 517,847 characters (with spaces) which is a quite voluminous and valuable resource for

budding researchers. A series of rigorous experiments is computed to make it evident that the best summarization results are achieved when the HHD corpus is divided into the ratio of 70:30 for training and testing purposes, respectively. In future analysis, the hold-out cross-validation can be extended to k-fold cross-validation [70].

Training Dataset: In the training phase, several generations of chromosomes are generated while keeping the highest fitness value at each generation and then compared among all the highest fitness. The best fitness value represents the suitable features for training the HHD corpus. At the end, sum up all the highest chromosomes of the training dataset divided by the number of chromosomes to gain the feature weight for the testing purpose.

Testing Dataset: In the testing phase, the rest of the HHD corpus- untouched during the training phase is considered. The test phase also undergoes pre-processing, feature extraction, and processing phases, respectively. The modification is in scoring each feature based upon the feature weights raised during the training process. Then, the remaining steps of the ATS process are executed, and the relevant summary is generated.

*4.2. Evaluation Metrics*

For evaluating the ATS process, the following four metrics are used: ROUGE, precision, recall, and F-measure.

Recall-Oriented Understudy for Gisting Evaluation (ROUGE) compares machine-generated summary (or system summary) with respect to human-generated summary (or reference summary). *ROUGE–N* measures n-gram recall or co-occurrences of n-grams which is calculated as is given in Equation (17).

$$ROUGE - N = \frac{\sum_{R_S \in (reference\ summaries)} \sum_{gram_{ng} \in R_S} Count_{match}(gram_{ng})}{\sum_{R_S \in (reference\ summaries)} \sum_{gram_{ng} \in R_S} Count(gram_{ng})} \tag{17}$$

where

$R_S$: reference sentence;
$Ng$: length of n-gram;
$Count(gram_{ng})$: total number of n-grams in reference sentence;
$Countmatch(gram_{ng})$: possible number of n-grams shared between system and reference sentence.

The precision metric is also called a positive predictive value, defined as a fraction of the relevant instances retrieved over the total of the retrieved instances. In other words, the precision metric in the context of ROUGE determines how much of the system summary is, in fact, relevant, as is given in Equation (18).

$$Precision = \frac{system\ summary \cap reference\ summary}{system\ summary} \tag{18}$$

Recall metric is also called sensitivity which is defined as a fraction of the relevant instances retrieved over the total of the relevant instances. In other words, recall metric in the context of ROUGE determines how much of the reference summary is the system summary recovering as is given in Equation (19).

$$Recall = \frac{system\ summary \cap reference\ summary}{reference\ summary} \tag{19}$$

As stated in Equations (18) and (19), system summary ∩ reference summary stands for the number of overlapping words between the system summary and reference summary [71]. An F-measure metric is also called an F-score which is defined in ROUGE as the harmonic mean of the precision and recall as is given in Equation (20).

$$F = \frac{2 * Precision * Recall}{Precision + Recall} \tag{20}$$

In this research, ROUGE-N (N = 1, 2), i.e., ROUGE-1 and ROUGE-2 metrics are taken into consideration. ROUGE-1 determines an overlap of unigrams between system and reference summaries. ROUGE-2 determines an overlap of bigrams between system and reference summaries.

### 4.3. Results and Discussion

This section discusses results related to feature weights, summary compression rates, ROUGE-N (N = 1, 2) evaluation, the time that is taken for the compressed summary, and comparison of summary among tools, respectively.

### 4.3.1. Feature Weights

In order to generate a high-quality summary, it is mandatory to study the impact of available features that are proposed in an ATS process. In this paper, the GA method for the Hindi text summarization task identifies which of the features are more important than others using calculated weights of the features.

Figure 3 shows the feature weights for different features. It is observed that the generated features—keywords ($F_{kw}$), sentence similarity ($F_{ss}$), named entities ($F_{ne}$), English–Hindi words ($F_{eh}$), and TF-ISF ($F_{ti}$)—have higher weights in comparison to other features—sentence paragraph position ($F_{sp}$), numerical data ($F_{nd}$), and length of sentence ($F_{ls}$). The highest feature weight computed is $F_{ss}$ (0.58) for the sentence similarity feature. Then, the following weight sequence is observed in decreasing order: $F_{ne}$ (0.52), $F_{ti}$ (0.47), $F_{kw}$ (0.45), $F_{eh}$ (0.41), $F_{sp}$ (0.31), and $F_{nd}$ (0.21), respectively. The weakest weight among all the features is $F_{ls}$ (0.12) for the length of the sentence feature.

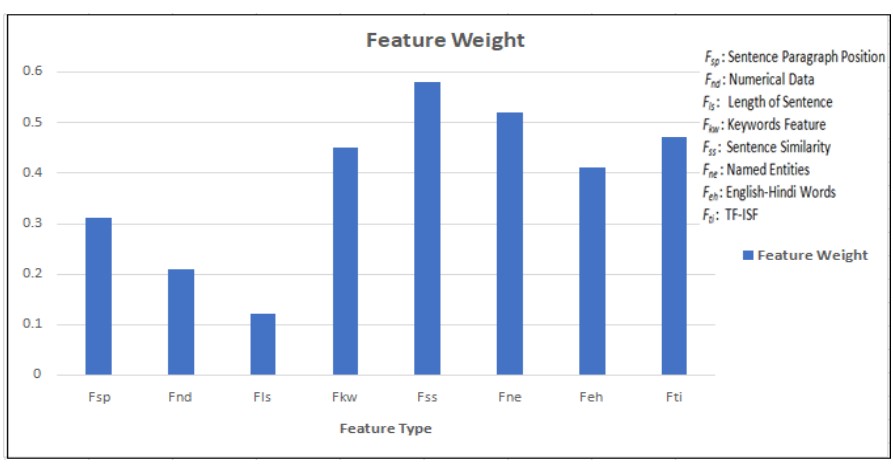

**Figure 3.** Feature weights for different features.

### 4.3.2. Summary of Compression Rates

In this research, various feature combinations are experimented over the HHD test dataset. The information on the best feature sets is discussed for different compression rates of 25%, 50%, and 65% (Tables 7–9), respectively.

Table 7 discusses the information on the best 14 feature sets (Set1 to Set 14) at 25% compression rate. For Set1, $F_{sp}$ alone gives F-measure as 0.52. For Set2, addition of $F_{nd}$ increases F-measure to 0.58. For Set3, adding $F_{ls}$ to $F_{sp}$ and $F_{nd}$ reduces the F-measure to 0.55 because of the lesser feature weight of $F_{ls}$. For Set4, involving $F_{kw}$ to $F_{sp}$ and $F_{nd}$ increases F-measure to 0.60. For Set5, addition of $F_{ss}$ to $F_{sp}$ and $F_{kw}$ increases F-measure to 0.64%. For Set6, consideration of the first four features $F_{sp}$, $F_{nd}$, $F_{ls}$, and $F_{kw}$ unalters the F-measure to 0.64. For Set7, two high-weighted features $F_{ss}$ and $F_{ne}$ increase the F-measure to 0.68. For Set8, inclusion of $F_{nd}$, $F_{kw}$, $F_{ss}$, and $F_{ne}$ further increases the F-measure by 0.72 since most of these features are high weighted features. For Set9, adding $F_{ls}$ to $F_{sp}$, $F_{kw}$, $F_{ss}$, and $F_{ne}$ again reduces the F-measure to 0.63. For Set10, consideration of $F_{kw}$, $F_{ss}$, $F_{ne}$, and $F_{eh}$ makes the F-measure 0.75. However, for Set11, the first five features and seventh feature

($F_{sp}$, $F_{nd}$, $F_{ls}$, $F_{kw}$, $F_{ss}$, and $F_{eh}$) again reduces the F-measure to 0.65 since some features are of lesser weights. For Set12, $F_{nd}$, $F_{kw}$, $F_{ss}$, $F_{eh}$, and $F_{ti}$ again increase F-measure as 0.75 since all other features except $F_{nd}$ are high weighted features. For Set13, including $F_{sp}$, $F_{kw}$, $F_{ss}$, $F_{ne}$, $F_{eh}$, and $F_{ti}$ further increases the F-measure to 0.78. For Set14, all eight features do not make any difference in the F-measure. Hence, Set13 and Set14 feature sets provide better results.

Table 8 discusses the information on the best 14 feature sets (Set1 to Set 14) at 50% compression rate. For Set1 and Set2, F-measure increases from 0.61 to 0.64. For Set3, adding $F_{ls}$ to $F_{sp}$ and $F_{nd}$ reduces F-measure to 0.62 because of the lesser feature weight of $F_{ls}$. For Set4 to Set6, the F-measure increases from 0.65 to 0.68. For Set7, Set8, and Set10, high-weighted features $F_{ss}$ and $F_{ne}$ increase the F-measure to 0.71, 0.72, and 0.74, respectively. However, for Set9 and Set11, $F_{ls}$ reduces the F-measure to 0.63 and 0.65. For Set12 and Set13, high-weighted features again increase the F-measure from 0.77 to 0.81. For Set14, all the features are incorporated to yield an F-measure of 0.83.

**Table 7.** Evaluation measure for 25% compression rate.

| Feature Set | Features | Precision | Recall | F-Measure |
| --- | --- | --- | --- | --- |
| Set1 | $F_{sp}$ | 0.54 | 0.51 | 0.52 |
| Set2 | $F_{sp}$, $F_{nd}$ | 0.64 | 0.53 | 0.58 |
| Set3 | $F_{sp}$, $F_{nd}$, $F_{ls}$ | 0.57 | 0.54 | 0.55 |
| Set4 | $F_{sp}$, $F_{nd}$, $F_{kw}$ | 0.62 | 0.59 | 0.60 |
| Set5 | $F_{sp}$, $F_{kw}$, $F_{ss}$ | 0.66 | 0.63 | 0.64 |
| Set6 | $F_{sp}$, $F_{nd}$, $F_{ls}$, $F_{kw}$ | 0.63 | 0.66 | 0.64 |
| Set7 | $F_{ss}$, $F_{ne}$ | 0.71 | 0.65 | 0.68 |
| Set8 | $F_{nd}$, $F_{kw}$, $F_{ss}$, $F_{ne}$ | 0.72 | 0.73 | 0.72 |
| Set9 | $F_{sp}$, $F_{ls}$, $F_{kw}$, $F_{ss}$, $F_{ne}$ | 0.63 | 0.64 | 0.63 |
| Set10 | $F_{kw}$, $F_{ss}$, $F_{ne}$, $F_{eh}$ | 0.74 | 0.76 | 0.75 |
| Set11 | $F_{sp}$, $F_{nd}$, $F_{ls}$, $F_{kw}$, $F_{ss}$, $F_{eh}$ | 0.69 | 0.62 | 0.65 |
| Set12 | $F_{nd}$, $F_{kw}$, $F_{ss}$, $F_{eh}$, $F_{ti}$ | 0.75 | 0.75 | 0.75 |
| Set13 | $F_{sp}$, $F_{kw}$, $F_{ss}$, $F_{ne}$, $F_{eh}$, $F_{ti}$ | 0.81 | 0.76 | 0.78 |
| Set14 | $F_{sp}$, $F_{nd}$, $F_{ls}$, $F_{kw}$, $F_{ss}$, $F_{ne}$, $F_{eh}$, $F_{ti}$ | 0.79 | 0.78 | 0.78 |

**Table 8.** Evaluation measure for 50% compression rate.

| Feature Set | Features | Precision | Recall | F-Measure |
| --- | --- | --- | --- | --- |
| Set1 | $F_{sp}$ | 0.62 | 0.61 | 0.61 |
| Set2 | $F_{sp}$, $F_{nd}$ | 0.63 | 0.66 | 0.64 |
| Set3 | $F_{sp}$, $F_{nd}$, $F_{ls}$ | 0.62 | 0.62 | 0.62 |
| Set4 | $F_{sp}$, $F_{nd}$, $F_{kw}$ | 0.65 | 0.66 | 0.65 |
| Set5 | $F_{sp}$, $F_{kw}$, $F_{ss}$ | 0.69 | 0.65 | 0.67 |
| Set6 | $F_{sp}$, $F_{nd}$, $F_{ls}$, $F_{kw}$ | 0.68 | 0.68 | 0.68 |
| Set7 | $F_{ss}$, $F_{ne}$ | 0.70 | 0.73 | 0.71 |
| Set8 | $F_{nd}$, $F_{kw}$, $F_{ss}$, $F_{ne}$ | 0.73 | 0.72 | 0.72 |
| Set9 | $F_{sp}$, $F_{ls}$, $F_{kw}$, $F_{ss}$, $F_{ne}$ | 0.64 | 0.63 | 0.63 |
| Set10 | $F_{kw}$, $F_{ss}$, $F_{ne}$, $F_{eh}$ | 0.73 | 0.75 | 0.74 |
| Set11 | $F_{sp}$, $F_{nd}$, $F_{ls}$, $F_{kw}$, $F_{ss}$, $F_{eh}$ | 0.62 | 0.69 | 0.65 |
| Set12 | $F_{nd}$, $F_{kw}$, $F_{ss}$, $F_{eh}$, $F_{ti}$ | 0.79 | 0.76 | 0.77 |
| Set13 | $F_{sp}$, $F_{kw}$, $F_{ss}$, $F_{ne}$, $F_{eh}$, $F_{ti}$ | 0.84 | 0.79 | 0.81 |
| Set14 | $F_{sp}$, $F_{nd}$, $F_{ls}$, $F_{kw}$, $F_{ss}$, $F_{ne}$, $F_{eh}$, $F_{ti}$ | 0.83 | 0.84 | 0.83 |

Table 9 discusses the information on the best 14 feature sets (Set1 to Set 14) at 65% compression rate. For Set1 and Set2, F-measure increases from 0.65 to 0.71. For Set3, adding $F_{ls}$ to $F_{sp}$ and $F_{nd}$ reduces the F-measure to 0.68 because of the lesser feature weight of $F_{ls}$. For Set4 to Set8, F-measure increases from 0.75 to 0.84 because of high-weighted features However, for Set9 and Set11, the F-measure reduces to 0.75 and 0.80, respectively, because of $F_{ls}$. For Set12 and Set13, high-weighted features increase the F-measure from 0.82 to 0.86. For Set14, all the features are incorporated to yield an F-measure of 0.87.

**Table 9.** Evaluation measure for 65% compression rate.

| Feature Set | Features | Precision | Recall | F-measure |
|---|---|---|---|---|
| Set1 | $F_{sp}$ | 0.62 | 0.69 | 0.65 |
| Set2 | $F_{sp}, F_{nd}$ | 0.69 | 0.74 | 0.71 |
| Set3 | $F_{sp}, F_{nd}, F_{ls}$ | 0.69 | 0.68 | 0.68 |
| Set4 | $F_{sp}, F_{nd}, F_{kw}$ | 0.73 | 0.78 | 0.75 |
| Set5 | $F_{sp}, F_{kw}, F_{ss}$ | 0.79 | 0.86 | 0.82 |
| Set6 | $F_{sp}, F_{nd}, F_{ls}, F_{kw}$ | 0.82 | 0.83 | 0.82 |
| Set7 | $F_{ss}, F_{ne}$ | 0.87 | 0.81 | 0.84 |
| Set8 | $F_{nd}, F_{kw}, F_{ss}, F_{ne}$ | 0.85 | 0.83 | 0.84 |
| Set9 | $F_{sp}, F_{ls}, F_{kw}, F_{ss}, F_{ne}$ | 0.72 | 0.79 | 0.75 |
| Set10 | $F_{kw}, F_{ss}, F_{ne}, F_{eh}$ | 0.84 | 0.86 | 0.85 |
| Set11 | $F_{sp}, F_{nd}, F_{ls}, F_{kw}, F_{ss}, F_{eh}$ | 0.76 | 0.84 | 0.80 |
| Set12 | $F_{nd}, F_{kw}, F_{ss}, F_{eh}, F_{ti}$ | 0.81 | 0.83 | 0.82 |
| Set13 | $F_{sp}, F_{kw}, F_{ss}, F_{ne}, F_{eh}, F_{ti}$ | 0.84 | 0.88 | 0.86 |
| Set14 | $F_{sp}, F_{nd}, F_{ls}, F_{kw}, F_{ss}, F_{ne}, F_{eh}, F_{ti}$ | 0.83 | 0.91 | 0.87 |

Observation of Tables 7–9, with an F-measure around 0.8, suggests that some elements are missing to better explain the pattern. These values match or exceed other published work for Hindi, detailed in Table 1, but nevertheless suggest that there is room for improvement. Although the present study compiles the most commonly used features in ATS, other features, which are not usually considered, could improve the quality of the summaries, such as semantic relations between terms, polysemy, coherence between sentences, or readability of the resulting summary.

In the case of Hindi, most of the studies do not provide information on the relative importance of each of the features. It is interesting to note that the least influential variable in the models is sentence length. This factor is highly correlated with readability, but has little impact on the accuracy of the summary [72]. Some works have suggested that the readability has a greater impact on the accuracy of the summaries [73]. The presence of numerical values, which the readability guidelines suggest replacing with textual quantifiers (e.g., greater than, similar, etc.), is also related to the readability recommendations. On the other hand, features that are traditionally more related to information retrieval because of their discriminatory value in relation to other documents have a higher weight, such as named entities, the presence of keywords, the frequency of terms, or the inverse frequency in the sentence. The number of English words in Hindi texts also has a clear discriminatory value, as the use of a foreign term is often due to the absence of the term or its synonyms in Hindi. As can be seen in Tables 7–9, the higher the compression ratio, the lower the F-measure. Another observation that seems obvious but needs clarification is that the more features the better the F-measure, even though there are sets, such as 7 that use only two features. This corroborates that even in elements that seem to be of lesser importance, such as sentence length, they also explain a part of the model not covered by other variables.

### 4.3.3. ROUGE-N Evaluation Measure

The ROUGE-N measure considers content overlap (counts all the shared words). It determines if the same concepts are pondered between the system and reference summaries. In other words, ROUGE does not assess how fluent the summary is; however, it tries to assess how adequate the summary is. ROUGE-N recall = 65% means that 65% of n-grams in the reference summary are also present in the system generated summary. Furthermore, ROUGE-N precision = 65% means that 65% of n-grams in the system generated summary are also present in the reference summary. Thus, ROUGE minimizes the need for human post-processing on the summary. Furthermore, in this research, the average performance of the system summary is 79% and 66%, approximately similar to the reference summary using ROUGE-1 and ROUGE-2, respectively (Table 10).

**Table 10.** ROUGE-N (N = 1, 2) evaluation measures.

|  | Average Precision | Average Recall | Average F-Measure |
|---|---|---|---|
| ROUGE-1 | 81% | 78% | 79% |
| ROUGE-2 | 65% | 68% | 66% |

It should be noted that Rouge has certain shortcomings, as aspects such as the variety of synonyms and related terms have a negative effect on the metric. These deficiencies are usually complemented by additional metrics that assess coherence or readability [73].

### 4.3.4. Time Comparison to Generate Summary

The graph (Figure 4) depicts the time taken by the proposed ATS methodology for generating the summary of the HHD corpus using varying compression rates. The system takes about 55 s to produce a 25% compression rate summary. Similarly, the system takes about 108 s to produce a 50% compression rate summary. Furthermore, the system takes about 119 s to produce a summary for a 65% compression rate. These time comparisons are displayed as a bar graph where the x-axis represents the compression rate and the y-axis represents the time in seconds, respectively.

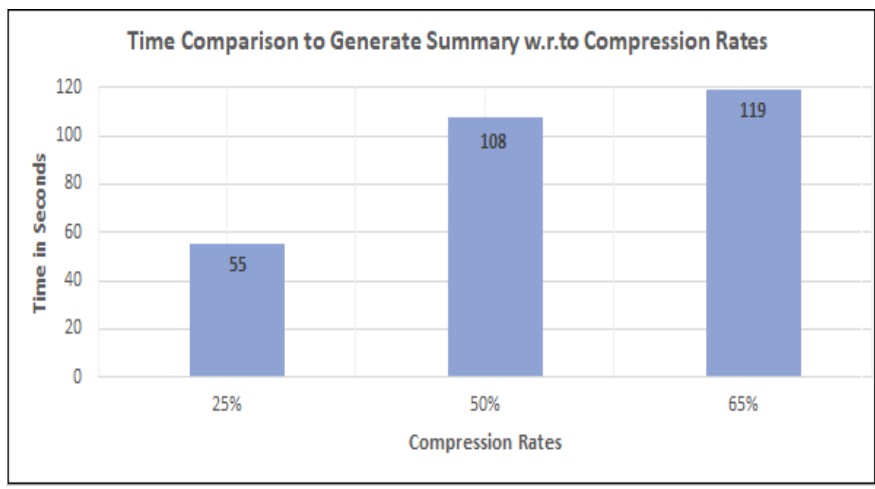

**Figure 4.** Time comparison to generate summary vs. comparison rates.

### 4.3.5. Comparison of Summary among Tools

Summary results are compared with the aforementioned tools, and it is found that the proposed work gives promising results with a good compression rate and efficiency (Table 11).

**Table 11.** Summary comparison among tools.

| Tools | Summary Reduction |
|---|---|
| Free Summarizer | 58% |
| Tool4Noobs | 66% |
| SMMRY | 63% |
| Proposed Work | 65% |

## 5. Conclusions

In this research, an Automatic Text Summarization (ATS) methodology for the Hindi language is proposed over the Hindi Health Data (HHD) corpus. ATS works with Real Coded Genetic Algorithm (RCGA) which optimizes the feature weights using selection, Simulated Binary Crossover (SBX), and Polynomial Mutation. RCGA selects the best chromosome which contains real-valued weights of generated features, computes the distance between sentence scores, and ranks the corpus sentences.

The experimentations are performed on different combinations of eight features. The distinguishing features among them are sentence similarity and named entity features which are combined with others for computing the evaluation metrics. The top 14 feature combinations are detailed over three compression rates—25%, 50%, and 65%. For 25% compression rate, precision (0.79), recall (0.78), and F-measure (0.78). For 50% compression rate, precision (0.83), recall (0.84), and F-measure (0.83). For 65% compression rate, precision (0.83), recall (0.91), and F-measure (0.87), respectively. The system takes about 55 s, 108 s, and 119 s to produce a summary for 25%, 50%, and 65% compression rates, respectively. The average performance of the system summary is ROUGE-1 (79%) and ROUGE-2 (66%) which is approximately similar to the reference summary. In comparison with other existing tools, the overall summary gives promising results with a reasonable compression rate and efficiency.

## 6. Future Work

In the future, the following research directions can be explored further:

- The automatic text summarization system can be applied to other domains such as finance, education, business, etc.
- Based upon the choice of domain, prominent features can be identified using RCGA.
- The ATS system can be portable to other Indian languages, such as Punjabi, Bengali, and Tamil.
- The hold-out cross-validation can be extended to k-fold cross-validation.
- The genetic algorithm optimized deep learning models, such as Recurrent Neural Network (RNN) and Long-Short Term Memory (LSTM), can be applied to improve the evaluation metrics.
- An interesting aspect to develop in the future is the impact of Rogue-L in Hindi, a metric that often does not appear in the ATS articles in this language. Already in Lin's article [74] the impact of some parameters on the computation of 17 Rogue-like indicators was compared showing the variation in English under different configurations. However, in languages where sentence order is more flexible or ellipsis are more frequent, an impact on this metric is to be expected. Thus, the aim is to evaluate Rogue-L under different configurations and languages to analyze its behavior.
- Additional features need to be taken into account in the model. It is a known fact that evaluation metrics do not always detect the accuracy of the summary, the use of terminological equivalences, readability or coherence are aspects to be included in a correct analysis and evaluation of ATS.

**Author Contributions:** The conceptualization, A.J. and K.V.K.; methodology, A.J. and A.A.; software, D.Y.; validation, J.M. and K.V.K.; formal analysis, A.J. and D.Y.; investigation, D.Y. and K.V.K.; resources, A.J. and A.A.; data curation, A.J.; writing—original draft preparation, A.J. and K.V.K.; writing—review and editing, A.A., D.Y., and J.M.; visualization, A.J. and D.Y.; supervision, A.A.; project administration, J.M. All authors have read and agreed to the published version of the manuscript.

**Funding:** This research received no external funding.

**Institutional Review Board Statement:** Not applicable.

**Informed Consent Statement:** Not applicable.

**Data Availability Statement:** The dataset which is considered in this research work is available freely at the Kaggle: Hindi Health Dataset (HHD) corpus. https://www.kaggle.com/datasets/aijain/hindi-health-dataset (accessed on 25 June 2021).

**Conflicts of Interest:** The authors declare no conflict of interest.

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
