# Peer review of "Automatic Text Summarization for Hindi Using Real Coded Genetic Algorithm"

_applsci, doi:10.3390/app12136584_

Round 1

Reviewer 1 Report

  • Paper Summary

The paper proposes an extractive text summarization method based on Real Coded Genetic Algorithm (RCGA) and applies it to health corpus in the Hindi language (Hindi Health Dataset). The proposed method (pipeline) consists of five stages of processing: pre-processing, feature extraction, processing, sentence ranking, and summary generation. Evaluation is performed using ROUGE-1 and ROUGE-2 scores. Three extractive summarization tools are used as baseline models against which they compare the proposed approach.  

  • Strengths

    • The paper proposes an extractive approach to the summarization of Hindi Language text (health corpus) using a real coded genetic algorithm. The application of real coded genetic algorithms to summarization is less explored and so the authors’ reuse of GA techniques for summarization seems novel.  

    • The authors provide a list of other natural language processing applications that can benefit from summarization which is not generally done in the literature. 

    • The study conducts a thorough investigation of the Hindi language addressing most of the nuances of the language; this is demonstrated in the preprocessing and feature extraction phases. 

    • The adoption of TF-ISF (Term Frequency Inverse Sentence Frequency), a sentence-level version of TF-IDF (which is a very popular technique), is interesting. 

    • The paper is easy to read but the English and the grammar can be improved. 

  •  

    • Weaknesses

      • Some background on Real Coded Genetic Algorithm (RCGA) would be helpful, in place of  verbatim repetitions of the steps of the proposed in multiple sections. While Genetic Algorithm (GA) for text summarization is discussed in Related work, RCGA is inadequately discussed in Processing. Further, all steps are “processing”-steps, so a descriptive label for this step such as a concise variant of “Genetic Algorithm-based Feature Composition and Processing for Summarization” would be helpful.

      • While the authors reported ROUGE-1, and ROUGE-2 scores, ROUGE-L (longest common subsequence) is missing. ROUGE-L is also an important metric to evaluate the quality of extractive summarization systems. 

      • Related Work (Section 2) seems more extensive than necessary (spans pages 2 - 8), while the description of contributions of the paper is adequate. Further, “abstractive summarization” reference in the “Future Work” seems dismissive while it is non-trivial and involved.

      •  
    • Writing Issues - Requires thorough and careful rewriting

    • L42: “On the other hand, the proliferation of information over the internet leads to manual textual summaries as a cumbersome task.” ⇒ What does this mean?

    • L40: “Inputted” => “input”

    • “club” => “combine”

    • L66: “Further, an additive advantage of the ATS approach through extractive summarization has played a pivotal role.”⇒ What does this mean?

    • L68 and others: “Processing”-step  => Must qualify it, because everything is processing.

    • Formatting issue at the bottom of Table 1. Decimal points are normally not broken onto a new line. 

    • L83-94: At the end of Introduction, the authors must clearly state and distinguish past work from the contributions of the current work. Specifically, novelty should be clearly distinguished from a reuse or adaptation of a technique.

    • L241: number of best words, keywords highlighting????

    • L267=268: DELETE

    • Section 3: Expand on RGCA. Include illustrative examples of similarity measures.

    • L: 313: such as A, B, C, etc.  ⇒  such as A, B, and C.        Or     A, B, C, etc.

    • L440-441: Rewrite.

    • L578: What is “intersection”?

    • Section 4.3: Table provides the evaluation metrics well. What is also needed in this section is a better description/interpretation of that information, not mere verbalization. That is, a detailed description of the implication of these results is what is expected here.

    • L630-633: 2 Missing references message.

    • The conclusion is repeating the processing rather than discussing the results/takeaway from the processing.

    • There are lots of grammatical errors throughout the paper,  too many to list. The authors must get proof read and update the paper thoroughly.

Author Response

Responses with respect to Reviewers' Comments

Manuscript Title: Automatic Text Summarization for Hindi Using Real Coded Genetic Algorithm

Manuscript ID: applsci-1713933

REVIEWER #1

Comment 1: Some background on Real Coded Genetic Algorithm (RCGA) would be helpful, in place of verbatim repetitions of the steps of proposed in multiple sections. While the Genetic Algorithm (GA) for text summarization is discussed in Related work, RCGA is inadequately discussed in Processing. Further, all steps are “processing”-steps, so a descriptive label for this step such as a concise variant of “Genetic Algorithm-based Feature Composition and Processing for Summarization” would be helpful.

Action: Thank you for the nice suggestion by the reviewer. Background on Real Coded Genetic Algorithm (RCGA) is added, in support of the comment by the reviewer.

Change Location: Section 1 Introduction, Fourth Paragraph, Highlighted by Green Colour; Section 2.2 Genetic Algorithm for Summarization, Last Paragraph, Highlighted by Green Colour; References [39-41] updated in References and cited within the text as well.

-------------------------------

Comment 2: While the authors reported ROUGE-1, and ROUGE-2 scores, ROUGE-L (longest common subsequence) is missing. ROUGE-L is also an important metric to evaluate the quality of extractive summarization systems. 

Action: Thank you for the nice suggestion by the reviewer. ROUGE-L evaluation metric requires rigorous experimentation which is too time-consuming at this stage. The reviewer is requested to consider it in the near future.

Change Location: Section 6, Future Work, Highlighted by Green Colour.

-------------------------------

Comment 3: Related Work (Section 2) seems more extensive than necessary (spans pages 2 - 8), while the description of contributions of the paper is adequate. Further, the “abstractive summarization” reference in the “Future Work” seems dismissive while it is non-trivial and involved.

Action: Thank you for the nice suggestion by the reviewer. Section 2, Related Work, extensive part is removed, now reduced to four sub-sections. Abstractive summarization from Future Work is removed, in support of the comment by the reviewer.

Change Location: Section 2, Related Work, Highlighted by Green Colour; Section 6, Future Work, Highlighted by Green Colour.

-------------------------------

Comment 4: Writing Issues - Requires thorough and careful rewriting

  • L42: “On the other hand, the proliferation of information over the internet leads to manual textual summaries as a cumbersome task.” ⇒ What does this mean?
  • L40: “Inputted” => “input”
  • “club” => “combine”
  • L66: “Further, an additive advantage of the ATS approach through extractive summarization has played a pivotal role.”⇒ What does this mean?
  • L68 and others: “Processing”-step  => Must qualify it, because everything is processing.
  • Formatting issue at the bottom of Table 1. Decimal points are normally not broken onto a new line. 
  • L83-94: At the end of the Introduction, the authors must clearly state and distinguish past work from the contributions of the current work. Specifically, novelty should be clearly distinguished from reuse or adaptation of a technique.
  • L241: number of best words, keywords highlighting????
  • L267=268: DELETE
  • L: 313: such as A, B, C, etc.  ⇒ such as A, B, and C. Or     A, B, C, etc.
  • L440-441: Rewrite.
  • L578: What is “intersection”?
  • L630-633: 2 Missing references message.

Action:

L42: Thank you for the nice suggestion by the reviewer. The statement is out of context at this place and hence is removed, in support of the comment by the reviewer.

L40: Thank you for the nice suggestion by the reviewer. The words are rectified, in support of the comment by the reviewer.

L66: Thank you for the nice suggestion by the reviewer. The statement is out of context at this place and hence is removed, in support of the comment by the reviewer.

L68: Thank you for the nice suggestion by the reviewer. The processing phase considers the Real Coded Genetic Algorithm (RCGA), whose search capability explores the best feature weights. The feature combinations are experimented with over the test dataset for different compression rates. The best-scored sentences are picked up and added to the final summary, and the generated summary is compared with other existing tools.

- Formatting issue at the bottom of Table 1 is resolved.

L83-94: Thank you for the nice suggestion by the reviewer. Research Contributions are mentioned, in support of the comment by the reviewer.

L241: Thank you for the nice suggestion by the reviewer. The statement is rectified, in support of the comment by the reviewer.

L267=268: Thank you for the nice suggestion by the reviewer. The statement is deleted, in support of the comment by the reviewer.

L313: Thank you for the nice suggestion by the reviewer. The statement is rectified, in support of the comment by the reviewer.

L440-441: Thank you for the nice suggestion by the reviewer. The statement is rewritten, in support of the comment by the reviewer.

L578: Thank you for the nice suggestion by the reviewer. The term “intersection” is explained, in support of the comment by the reviewer.

L630-633: Thank you for the nice suggestion by the reviewer. Table references are marked, in support of the comment by the reviewer.

Change Location:

L42: Section 1, Introduction, Highlighted by Green Colour

L40: Section 1, Introduction, Highlighted by Green Colour

L66: Section 1, Introduction, Highlighted by Green Colour

L68: Section 1, Introduction, last few lines of the Fourth Paragraph; Bottom of Table 1 is highlighted by Green Colour

L83-94: Section 1, Introduction, Second Last Paragraph, Highlighted by Green Colour

L241: Section 2.4, Tools for Text Summarization, Highlighted by Green Colour

L267: Section 3, Proposed Methodology, Second Paragraph

L313: Section 3.1.5, Stop-words Removal, Highlighted by Green Colour

L440-441: Section 3.3.1, Genetic Algorithm, Highlighted by Green Colour

L578: Section 4.2, Evaluation Metrics, Highlighted by Green Colour

L630-633: Section 4.3.2, Summary of Compression Rates, Table 7 to Table 9 are highlighted in Green Colour.

-------------------------------

Comment 5: Section 3: Expand on RGCA. Include illustrative examples of similarity measures.

Action: Thank you for the nice suggestion by the reviewer. RCGA is extended, in support of the comment by the reviewer.

Change Location: Section 3.3.1, Genetic Algorithm, Highlighted by Green Colour in Crossover and Mutation Subsections, Table 5 and Table 6.

-------------------------------

Comment 6: Section 4.3: Table provides the evaluation metrics well. What is also needed in this section is a better description/interpretation of that information, not mere verbalization. That is, a detailed description of the implication of these results is what is expected here.

Action: Thank you for the nice suggestion by the reviewer. A detailed description of the implication of these results is added, in support of the comment by the reviewer.

Change Location: Section 4.3.2, Summary of Compression Rates, Table 7 to Table 9 are highlighted in Green Colour.

-------------------------------

Comment 7: The conclusion is repeating the processing rather than discussing the results/takeaway from the processing.

Action: Thank you for the nice suggestion by the reviewer. The conclusion is rewritten, in support of the comment by the reviewer.

Change Location: Section 5, Conclusions, Highlighted by Green Colour.

-------------------------------

Comment 8: There are lots of grammatical errors throughout the paper, too many to list. The authors must get proofread and update the paper thoroughly.

Action: Thank you for the nice suggestion by the reviewer. The paper is proofread and updated, in support of the comment by the reviewer.

Change Location: Throughout the paper.

-------------------------------

Reviewer 2 Report

A real coded genetic algorithm is proposed to address the automatic text summarization problem for Hindi. In general, this paper is well organized with detailed literature review, clear problem formulations, and experimental analysis. A major revision is recommended. Please refer to my comments as follows.

Comment 1. Abstract:

(a) Define the acronym ROUGE.

(b) Clarify the distinguishing features and if they are differed to 14 features.

(c) Briefly highlight the improvement by proposed work, compared with existing works.

Comment 2. Include more terms to better reflect the scopes of the paper.

Comment 3. Section I Introduction:

(a) Elaborate the importance of ATS.

(b) Consider to update the acronym to a more conventional one as TF-IDF.

(c) The research contributions should be more specific to explain how the proposal addresses the limitations of the existing works and new research contents are conducted.

Comment 4. Section II Related Works:

(a) Elaborate the three types of extraction methods with more references.

(b) Justify the selection of GA instead of a broad coverage that includes other optimization algorithms.

(c) Summarize the limitations of existing works.

Comment 5. Section III Proposed methodology:

(a) Elaborate the formulations.

(b) Elaborate Table 4.

Comment 6. Section IV Experimental Setup:

(a) Authors are adopting a hold-out cross-validation with a ratio of 70:30 for training and testing dataset. It is suggested to cite the following article (title as follows) as a remark of future analysis that could be extended to a k-fold cross-validation:

- A Genetic Algorithm Optimized RNN-LSTM Model for Remaining Useful Life Prediction of Turbofan Engine

(b) Based on Tables 5-7, what are the findings and trends of results in varying the compression rates?

Comment 7. Conclusion, authors may elaborate the key findings to emphasize the research contributions.

Comment 8. Elaborate future research directions.

Author Response

Responses with respect to Reviewers' Comments

Manuscript Title: Automatic Text Summarization for Hindi Using Real Coded Genetic Algorithm

Manuscript ID: applsci-1713933

REVIEWER #2

Comment 1: Comment 1. Abstract:

(a) Define the acronym ROUGE.

(b) Clarify the distinguishing features and if they are differed to 14 features.

(c) Briefly highlight the improvement by proposed work, compared with existing works.

Action: Thank you for the nice suggestion by the reviewer. The abstract is rectified, in support of the comment by the reviewer.

Change Location: Abstract is highlighted in Green Colour while including Comments (a) to (c).

-------------------------------

Comment 2: Comment 2. Include more terms to better reflect the scopes of the paper.

Action: Thank you for the nice suggestion by the reviewer. More terms are included, in support of the comment by the reviewer.

Change Location: Keywords are added further, highlighted by Green Colour.

-------------------------------

Comment 3: Comment 3. Section I Introduction:

(a) Elaborate the importance of ATS.

(b) Consider to update the acronym to a more conventional one as TF-IDF.

(c) The research contributions should be more specific to explain how the proposal addresses the limitations of the existing works and new research contents are conducted.

Action: Thank you for the nice suggestion by the reviewer. Importance of ATS is elaborated. More conventional acronyms are used. Research contributions are rewritten, in support of the comment by the reviewer.

Change Location: Section 1, Introduction, First Paragraph, Highlighted by Green Colour with cited References [66] and [67]. Section 1, Introduction, Third Paragraph, Highlighted the Acronyms by Green Colour. Section 1, Introduction, Second Last Paragraph, Highlighted the Research Contributions by Green Colour.

-------------------------------

Comment 4: Section II Related Works:

(a) Elaborate the three types of extraction methods with more references.

(b) Justify the selection of GA instead of a broad coverage that includes other optimization algorithms.         

(c) Summarize the limitations of existing works.

Action: Thank you for the nice suggestion by the reviewer. The three types of extraction methods are elaborated with more references, in support of the comment by the reviewer.

Thank you for the nice suggestion by the reviewer. The selection of GA is justified, in support of the comment by the reviewer.

Thank you for the nice suggestion by the reviewer. The limitation of existing works is added, in support of the comment by the reviewer.

Change Location: Section 2.1, Types of Extractive Methods, Highlighted by Green Colour.

Section 1, Introduction, Fourth Paragraph, Highlighted by Green Colour.

Section 2.2, Genetic Algorithm for Summarization, Last Paragraph, Highlighted by Green Colour.

-------------------------------

Comment 5: Section III Proposed methodology:

(a) Elaborate the formulations.

(b) Elaborate Table 4.

Action: Thank you for the nice suggestion by the reviewer. The formulations are elaborated, in support of the comment by the reviewer.

Thank you for the nice suggestion by the reviewer. Table 4 is elaborated, in support of the comment by the reviewer.

Change Location: Section 3.3.1, Genetic Algorithm, Table 4, Highlighted by Green Colour.

Section 3.2.3, Length of Sentence; Section 3.2.6, Named Entities within Sentence; Section 3.2.7, English Hindi Words within Sentence; Section 3.3.1, Genetic Algorithm, Subsections- Selection of Best Chromosome, Crossover and Mutation, all Highlighted by Green Colour.

-------------------------------

Comment 6: Section IV Experimental Setup:

(a) Authors are adopting a hold-out cross-validation with a ratio of 70:30 for training and testing dataset. It is suggested to cite the following article (title as follows) as a remark of future analysis that could be extended to a k-fold cross-validation:

- A Genetic Algorithm Optimized RNN-LSTM Model for Remaining Useful Life Prediction of Turbofan Engine

(b) Based on Tables 5-7, what are the findings and trends of results in varying the compression rates?

Action: Thank you for the nice suggestion by the reviewer. In future analysis, k-fold cross-validation is extended and the suggested paper is cited, in support of the comment by the reviewer.

Thank you for the nice suggestion by the reviewer. Findings and trends of results in varying compression rates are added, in support of the comment by the reviewer.

Change Location: Section 6, Future Work, paper is cited as [68], Highlighted by Green Colour.

Section 4.3.2, Summary of Compression Rates, Table 7-9, Highlighted by Green Colour.

-------------------------------

Comment 7: Conclusion, authors may elaborate the key findings to emphasize the research contributions.

Action: Thank you for the nice suggestion by the reviewer. The conclusion is rewritten.

Change Location: Section 5, Conclusions, Highlighted by Green Colour.

-------------------------------

Comment 8: Elaborate on future research directions.

Action: Thank you for the nice suggestion by the reviewer. The future research directions are elaborated, in support of the comment by the reviewer.

Change Location: Section 6, Future Work, Highlighted by Green Colour.

Round 2

Reviewer 1 Report

We have looked through the author's response and updates. There are still lingering typos and grammatical errors even in the green highlighted parts, and not all comments have been satisfactorily addressed. Here are several examples of that. (1) The current version of the manuscript continues to verbalize the table of results, which we suggested they eliminate or at least enhance. Instead, we had requested an interpretation of the results of their experimentation, which would have provided more insights over and above what is in the table. (2) For ROUGE-L evaluation,  the authors responded that rigorous experiments would have to be conducted and that it could be time-consuming to do so. We don't think ROUGE-L evaluation would be time-consuming as we did not ask for new experiments. Without ROUGE-L, the evaluation would be incomplete.  (3) In some situations where we asked the authors to expand on, the authors have simply deleted those sentences. We are unsure if the authors understood our comments.   While the paper still needs improvement, we are leaning on weak accept only because it presents an application of GA to summarization and the details of working with the nuances of the Hindi language, both of which are novel or at least unconventional.   

Author Response

Responses with respect to Reviewers' Comments

Manuscript Title: Automatic Text Summarization for Hindi Using Real Coded Genetic Algorithm

Manuscript ID: applsci-1713933

REVIEWER #1

Comment 1: The current version of the manuscript continues to verbalize the table of results, which we suggested they eliminate or at least enhance. Instead, we had requested an interpretation of the results of their experimentation, which would have provided more insights over and above what is in the table.  (Section 4.3)

Action: Thank you for your suggestion, different comments interpreting the results obtained have been added and marked in orange. Sections: 4.3.2. Summary of Compression Rates, 4.3.3. ROUGE-N Evaluation Measure

-------------------------------

Comment 2: For ROUGE-L evaluation, the authors responded that rigorous experiments would have to be conducted and that it could be time-consuming to do so. We don't think ROUGE-L evaluation would be time-consuming as we did not ask for new experiments. Without ROUGE-L, the evaluation would be incomplete.

Answer: Thank you for your suggestion, we agree that it is a useful metric but that its usefulness, without further research, can be misinterpreted. As we have indicated our intention is to include it in future research, but we believe it is important to study it under different configurations (stopwords and stemming within additional data, such as readability and coherence). We believe that the inclusion of Rogue-L should be done with this additional information for a correct interpretation of the results. It is well known that Rogue metrics are not exempt from criticism (e.g., see the work of Tay, 2019, on "Suitability of Rouge for Opinion Summary Evaluation" or the most cited work of Lin, 2004). Space limitations have not allowed us to include these aspects in this article. It should be noted, in this regard, that the main reason for including Rouge-1 and Rouge-2 is to allow an objective comparison with other Hindi summary methods, where the Rogue-L metric is not usually included. The reason for the popularity of Rogue-1 and 2 is based on the study of the most cited article on Rogue by Dr. Lin ("A package for automatic evaluation of summaries", 2004). Lin compared 17 measures of Rogue. His study indicates that Rogue-1 and Rogue-2 are good indicators, regardless of whether they are summaries of a document or a set of documents, a feature on which Rogue-L does not excel. Unfortunately, very few publications include Rouge-L for Hindi; therefore, it is not as valuable a metric in this regard as the commonly included Rogue-1 and Rogue-2. It can be argued that it has comparative utility with other languages. Even so, this indicator is very sensitive to language characteristics, as in the case of English, where Rouge-L tends to differ significantly because flexibility in sentence construction, ellipsis and cohesive elements are different between languages, so the comparison may be more misleading than the other metrics.

Action: Future research on Rogue-L detailed in the text. Highlighted in orange. Sections: 4.3.3. ROUGE-N Evaluation Measure and 6. Future Work

Comment 3: There are still lingering typos and grammatical errors even in the green highlighted parts

Action: Some typos localized and corrected. Highlighted in orange

Reviewer 2 Report

Authors have significantly improved the quality of the manuscript. I recommend to accept the paper with minor revision (minor follow-up comments).

Comment 5: Section III Proposed methodology:

 (b) Elaborate Table 4.

Action: Thank you for the nice suggestion by the reviewer. The formulations are elaborated, in support of the comment by the reviewer.

Thank you for the nice suggestion by the reviewer. Table 4 is elaborated, in support of the comment by the reviewer.

Change Location: Section 3.3.1, Genetic Algorithm, Table 4, Highlighted by Green Colour.

Section 3.2.3, Length of Sentence; Section 3.2.6, Named Entities within Sentence; Section 3.2.7, English Hindi Words within Sentence; Section 3.3.1, Genetic Algorithm, Subsections- Selection of Best Chromosome, Crossover and Mutation, all Highlighted by Green Colour.

Follow-up comment: Authors may clarify if the contents showing in the table are example or the result of the intermediate step of the algorithm.

-------------------------------

Comment 8: Elaborate on future research directions.

Action: Thank you for the nice suggestion by the reviewer. The future research directions are elaborated, in support of the comment by the reviewer.

Change Location: Section 6, Future Work, Highlighted by Green Colour.

Follow-up comment: Authors may consider to elaborate the prominent features and the ROUGE-L evaluation metric.

Author Response

Responses with respect to Reviewers' Comments

Manuscript Title: Automatic Text Summarization for Hindi Using Real Coded Genetic Algorithm

Manuscript ID: applsci-1713933

REVIEWER #2

Comment 1:  (Table 4) Authors may clarify if the contents showing in the table are examples or the result of the intermediate step of the algorithm.

Answer:

This is an example chromosome from the initial population with the most relevant features in the literature shown in Table 1, along with the initial parameter values. We then proceed to the selection of chromosomes based on the fitness function.

Action:

Text added, highlighted in orange

-------------------------------

Comment 2: Elaborate on future research directions. Authors may consider elaborating the prominent features and the ROUGE-L evaluation metric.

Answer:

Thank you for your suggestion.

An interesting aspect of future development is the impact of Rogue-L in Hindi, a metric that often does not appear in the ATS articles in this language. In Lin's article [71], the impact of some parameters on the computation of 17 Rogue-like indicators was compared, showing the variation in English under different configurations. But in languages where sentence order is more flexible or ellipsis is more frequent, an impact on this metric is expected. Thus, the aim is to evaluate Rogue-L under different configurations and languages to analyze its behavior.

Although the inclusion of the most frequent indicators in the literature in the study has shown promising results, other characteristics should be considered. It is a well-known fact that evaluation metrics do not always detect the accuracy of the summary. The use of terminological equivalences, the polarity of the sentence in opinion analysis, the summary readability or its coherence are aspects to be included in a correct analysis and evaluation of ATS.

Action:

Text added, highlighted in orange. Sections: 4.3.2. Summary of Compression Rates, 4.3.3. ROUGE-N Evaluation Measure, and 6. Future Work

Reference [71] included

-------------------------------
